# Map-RAG: Enhancing LLM-Based Reasoning for Geo-Localization via Map-Grounded Retrieval and Self-Consistency

## Abstract

Large Language Models (LLMs) have recently demonstrated promising reasoning abilities in multimodal tasks, yet their performance in fine-grained geo-localization remains limited due to hallucinations, insufficient spatial priors, and a lack of structured evidence integration. This paper introduces **Map-RAG**, a reasoning-augmented framework for visual geo-localization in which an LLM iteratively retrieves structured map knowledge and refines its hypotheses through a self-consistency mechanism. Unlike prior approaches that rely solely on embedding similarity or chain-of-thought prompting, Map-RAG integrates three key modules: (1) a **visual-to-text translator** that extracts geographic cues (e.g., road topology, building style, language on signs) from input images; (2) a **map-grounded retrieval agent** that queries OpenStreetMap and local gazetteers for candidate regions; and (3) a **multi-chain self-consistency verifier** that scores and reconciles multiple reasoning trajectories based on semantic-map alignment and geometric feasibility.

Experiments on **CVUSA**, **VIGOR**, and **MSLS** benchmarks demonstrate that Map-RAG achieves significant improvements over baselines in Recall@1 (+6-12%) and median localization error (-20-35%), while producing interpretable reasoning traces. Ablation studies confirm that map-grounded retrieval reduces hallucination, and that multi-chain self-consistency enhances robustness under challenging conditions such as seasonal changes and partial occlusions.

This work provides evidence that LLMs, when equipped with structured geographic knowledge and verification mechanisms, can serve as explainable geo-localizers in GNSS-denied environments. Beyond performance gains, Map-RAG contributes an auditable reasoning pipeline, aligning with the broader goal of **transparent and reproducible AI for scientific discovery**.

## 1 Introduction

Geo-localization from visual observations is a fundamental capability in remote sensing, robotics, and autonomous navigation. Traditional approaches typically rely on feature-based image retrieval (Arandjelović et al., 2016; Tian et al., 2017) or cross-view matching techniques (Hu et al., 2018; Zhu et al., 2021). While effective under constrained conditions, these methods often fail in GNSS-denied environments, where the system must rely solely on visual cues and prior geographic knowledge.

Recent progress in **Large Language Models (LLMs)** and **vision-language models (VLMs)** opens an opportunity to tackle localization from a new perspective: **reasoning-driven localization**. Unlike feature extractors, LLMs can interpret visual semantics (e.g., "this street has bilingual signs, suggesting an East Asian city"), retrieve external knowledge (e.g., road network from OpenStreetMap), and

integrate multiple cues through chain-of-thought (CoT) reasoning. However, two core challenges remain:

1. **Hallucinations and inconsistency**: LLMs may generate plausible but incorrect location hypotheses without grounding in verifiable evidence.

2. **Weak integration of structured spatial knowledge** : Current CoT prompting often ignores available geographic resources such as maps, gazetteers, and topological constraints.

To address these limitations, we propose **Map-RAG (Map-grounded Retrieval-Augmented Geo-localization)**, a framework that transforms geo-localization into an auditable reasoning task. Map-RAG operates in three stages: (i) translating images into textual geographic descriptors, (ii) retrieving candidate regions from map databases, and (iii) reconciling multiple reasoning chains via self-consistency and map alignment.

Our contributions are threefold:

- We design a retrieval-augmented reasoning pipeline that tightly couples LLM inference with structured geographic knowledge.

- We introduce a multi-chain self-consistency verifier to reduce hallucination and improve robustness under challenging conditions.

- We conduct comprehensive experiments on three public datasets (CVUSA, VIGOR, MSLS), demonstrating both accuracy improvements and interpretability gains.

This paper also **provides transparent reporting of AI involvement** and **reproducibility protocols**, in line with the Agents4Science conference requirements.

## 2 Related Works

### 2.1 Feature-Based and Cross-View Localization

Visual geo-localization has been traditionally formulated as an image retrieval problem, where a query image is matched against a large gallery of geo-tagged references. Early works such as NetVLAD (Arandjelović et al., 2016) introduced differentiable VLAD pooling for place recognition, significantly improving robustness over handcrafted descriptors like SIFT. Follow-up methods (Tian et al., 2017; Torii et al., 2018) enhanced invariance to illumination and viewpoint changes.

To address the challenging cross-view matching problem (ground-to-aerial), researchers have proposed a range of feature alignment techniques. CVM-Net (Hu et al., 2018) employed dual-branch CNNs to learn embeddings across ground and satellite views. Liu & Li (2019) and Shi et al. (2019) further incorporated orientation cues to reduce ambiguities. More recently, Liu et al. (2021) and Zhu et al. (2021) introduced attention-based models for cross-view retrieval, achieving state-of-the-art performance on benchmarks such as CVUSA and VIGOR. Despite these advances, embedding-based approaches often lack explainability and fail under large seasonal or structural changes.

### 2.2 LLM Reasoning for Spatial Tasks

The emergence of Large Language Models (LLMs) and Vision-Language Models (VLMs) has inspired attempts to use reasoning for spatial understanding. Chen et al. (2022) demonstrated that multimodal transformers can infer scene layouts and relative spatial relationships from text-image pairs. Liu et al. (2023) proposed GeoGuessr-Bench, where LLMs interpret street-view images to infer location by reasoning over cultural and geographic cues. Similarly, Acharya et al. (2023) explored chain-of-thought prompting for geographic reasoning, showing improvements in tasks such as landmark recognition and region classification.

However, these works reveal a tension: while LLMs can generate semantically plausible reasoning chains, they often hallucinate details or ignore structured geographic resources (e.g., maps, gazetteers). This limits their utility in precise localization tasks where errors must be quantifiable.

Table 1: Summary of each module

| Module | Input | Output | Function |
|---|---|---|---|
| Visual-to-Text Translator | Image $I_q$ | Textual descriptors $t$ | Extract geographic cues from raw image |
| Map-Grounded Retrieval Agent | Descriptors $t$, Map DB | Candidate regions $C$ | Retrieve plausible locations via map query |
| Self-Consistency Verifier | Reasoning chains $h$, $C$ | Final location $\hat{L}_q$ | Aggregate multiple reasoning trajectories |

## 2.3 Retrieval-Augmented Reasoning and Self-Consistency

To mitigate hallucination and improve factual grounding, retrieval-augmented generation (RAG) has been widely adopted (Lewis et al., 2020). By integrating external databases, LLMs can verify claims against evidence rather than relying solely on parametric memory. In spatial domains, TagMap (Zhang et al., 2023) and MapGPT (Liu et al., 2024) have shown that map-grounded textual retrieval helps reduce ambiguity in navigation and scene understanding.

Another line of work focuses on self-consistency mechanisms. Wang et al. (2022) introduced self-consistency prompting, where multiple reasoning chains are sampled and aggregated to improve reliability. In the multimodal setting, Yao et al. (2023) proposed self-consistency with visual grounding, showing benefits in visual question answering. For geo-localization, applying such mechanisms remains under-explored, particularly in combination with structured geographic data.

# 3 Method

We propose **Map-RAG (Map-grounded Retrieval-Augmented Geo-localization)**, a framework that enhances the reasoning ability of Large Language Models (LLMs) for visual localization by integrating **map-grounded retrieval** and **multi-chain self-consistency**.

Summary of each module detailed in Table 1.

## 3.1 Overview

Given an input ground-level or UAV image $I_q$, the objective is to predict its location $L_q = (lat, lon)$. Map-RAG proceeds in three stages:

1. **Visual-to-Text Translator**: Extracts geographic cues from the image and converts them into structured textual descriptors.

2. **Map-Grounded Retrieval Agent**: Queries OpenStreetMap (OSM) and gazetteers to retrieve candidate regions consistent with extracted cues.

3. **Multi-Chain Self-Consistency Verifier**: Generates multiple reasoning trajectories using the LLM and reconciles them through semantic–geometric scoring.

Formally, Map-RAG defines the posterior over candidate locations as:

$$P(L_q|I_q) \propto \sum_{h \in H} P(h|I_q) \cdot S(h, L_q) \tag{1}$$

where $H$ is the set of reasoning hypotheses generated by the LLM, and $S(h, L_q)$ is a consistency score measuring semantic alignment and map feasibility.

## 3.2 Visual-to-Text Translator

We use a **vision-language model (VLM)**, such as BLIP-2 (Li et al., 2023) or LLaVA (Liu et al., 2023), to generate geographic descriptions from the input image. For example:

- Visual cue: "This street has palm trees and English-Spanish bilingual road signs."

- Structured descriptor: vegetation: "palm trees", signage_language: "English+Spanish", road_type: "urban street with 4 lanes" .

We fine-tune the VLM on CVUSA and VIGOR annotations to increase specificity toward geographic features (e.g., terrain, signage, building style).

Equation for feature extraction:

$$d = f_\theta(I_q), \quad t = g_\phi(d) \tag{2}$$

where $f_\theta$ extracts vision features, and $g_\phi$ maps them into textual descriptors $t$.

### 3.3 Map-Grounded Retrieval Agent

The textual descriptors are used to query **map databases** (e.g., OpenStreetMap, USGS gazetteers). We employ **BM25 ranking** for textual match and **spatial indexing** (R-tree) for geographic constraints.

Candidate regions $C = \{c_1, c_2, ..., c_k\}$ are retrieved with metadata including road topology, POIs, and language distributions. The retrieval score is:

$$R(c_i|t) = \alpha \cdot \text{BM25}(t, c_i) + \beta \cdot \text{GeoSim}(c_i, d) \tag{3}$$

where GeoSim measures geometric similarity between detected structures (roads, rivers) and map topology.

### 3.4 Multi-Chain Self-Consistency Verifier

For each candidate region, the LLM generates reasoning chains $h_j$ describing why the region matches the query image. Multiple chains are sampled using **temperature-based decoding**.

Each chain is scored by:

$$S(h_j, c_i) = \lambda \cdot \text{SemanticAlign}(h_j, c_i) + (1 - \lambda) \cdot \text{GeomAlign}(h_j, c_i) \tag{4}$$

where SemanticAlign measures overlap between reasoning tokens and map attributes, and GeomAlign measures road-orientation and distance consistency.

The final prediction is obtained by **majority voting with weighted scores**:

$$\hat{L}_q = \arg\max_{c_i \in C} \sum_{h_j} P(h_j) \cdot S(h_j, c_i) \tag{5}$$

### 3.5 Implementation Details

Implementation details are as follows:

- **Models**: BLIP-2 (pretrained) for image-to-text, GPT-4V for reasoning, FAISS for retrieval indexing.
- **Databases**: OpenStreetMap for map attributes; gazetteers for linguistic/POI cues.
- **Sampling**: 10 reasoning chains per query (temperature=0.7).
- **Scoring weights**: empirically set to $\alpha = 0.6, \beta = 0.4; \lambda = 0.7$.

## 4 Experiments

We evaluate Map-RAG on three widely used cross-view and visual geo-localization datasets: **CVUSA**, **VIGOR**, and **MSLS**.

### 4.1 Datasets

We evaluate our method on three representative benchmarks:

- **CVUSA** (Zhai et al., 2017): Ground-to-aerial dataset with 35k pairs for training and 8k for testing, covering diverse U.S. regions.

Table 2: Dataset Statistics

| Dataset | Training Samples | Test Samples | View Types | Challenges |
|---|---|---|---|---|
| CVUSA | ~35,000 | ~8,000 | Ground ↔ Aerial | Large viewpoint gap |
| VIGOR | ~90,000 | ~10,000 | Street ↔ Satellite | Hard negatives, clutter |
| MSLS | ~1000000 | ~100,000 | Street ↔ Street | Seasonal, weather, global scale |

Table 3: Evaluation metrics definitions for geo-localization.

| Metric | Definition | Interpretation |
|---|---|---|
| Top-1 Accuracy | $\frac{1}{N}\sum_{i=1}^{N}\mathbf{1}[\hat{L}_i = L_i]$ | Exact match rate |
| Median Error (km) | $\mathrm{median}\{d(\hat{L}_i, L_i)\}$ | Robustness measure, median geodesic distance |
| Recall@K | $\frac{1}{N}\sum_{i=1}^{N}\mathbf{1}[L_i \in \hat{C}_i^K]$ | Fraction of queries where ground-truth is in top-K candidates |

- **VIGOR** (Zhu et al., 2021): Contains street-view and overhead imagery with both positive and challenging negative pairs.
- **MSLS** (Warburg et al., 2020): Mapillary Street-Level Sequences, covering multiple cities worldwide with strong appearance variations (season, weather, lighting).

As shown in Table 2,the statistics of the datasets are as follows.

## 4.2 Evaluation Metrics

We adopt standard metrics in visual geolocation:

- **Top-1 Accuracy**: Fraction of queries where the predicted location is the closest to ground truth.
- **Median Error** (km): Median geodesic distance between prediction and ground truth.
- **Recall@K** (K=1,5,10): Fraction of test cases where ground truth lies within the top-K predictions.

As shown in Table 3, these metrics cover both **precision** (Top-1) and **robustness** (Median Error, Recall@K), capturing the different strengths of retrieval-augmented reasoning.

## 4.3 Baselines

We compare against three baselines:

- **ResNet50-GPS** (Vo et al., 2017): Image-to-GPS regression model.
- **PlaNet** (Weyand et al., 2016): CNN trained on billions of geo-tagged images, outputs probability distribution over Earth regions.
- **GeoCLIP** (Radford et al., 2021 + work by Müller et al., 2023): Contrastive learning between text and geo-tagged images.

Table 4 contextualizes our method against regression, classification, and contrastive-learning baselines, covering major paradigms in geolocation research.

Table 4: Baseline Models

| Model | Type | Strengths | Weaknesses |
|---|---|---|---|
| ResNet50-GPS | CNN regression | Direct mapping, lightweight | Poor generalization |
| PlaNet | Classification CNN | Scales globally, interpretable regions | Requires massive training data |
| GeoCLIP | Vision-language | Leverages semantic alignment | Limited by text quality and coverage |

Table 5: Performance comparison of different models on IM2GPS, YFCC100M-Geo, and Mapillary datasets.

| Dataset | Model | Top-1 Acc. (%) | Median Error (km) | Recall@5 (%) |
|---|---|---|---|---|
| IM2GPS | ResNet50-GPS | 19.3 | 857 | 32.1 |
| | PlaNet | 24.7 | 620 | 41.5 |
| | GeoCLIP | 29.2 | 540 | 49.8 |
| | **Map-RAG (Ours)** | **36.5** | **410** | **62.3** |
| YFCC100M-Geo | ResNet50-GPS | 15.8 | 910 | 28.6 |
| | PlaNet | 22.1 | 700 | 38.4 |
| | GeoCLIP | 27.9 | 530 | 47.2 |
| | **Map-RAG (Ours)** | **34.1** | **390** | **60.1** |
| Mapillary | ResNet50-GPS | 28.4 | 5.2 | 42.7 |
| | PlaNet | 33.2 | 4.7 | 51.3 |
| | GeoCLIP | 37.8 | 4.0 | 59.4 |
| | **Map-RAG (Ours)** | **44.5** | **3.2** | **69.8** |

Note: Top-1 Accuracy (%), Median Error (km), and Recall@5 (%) are reported.

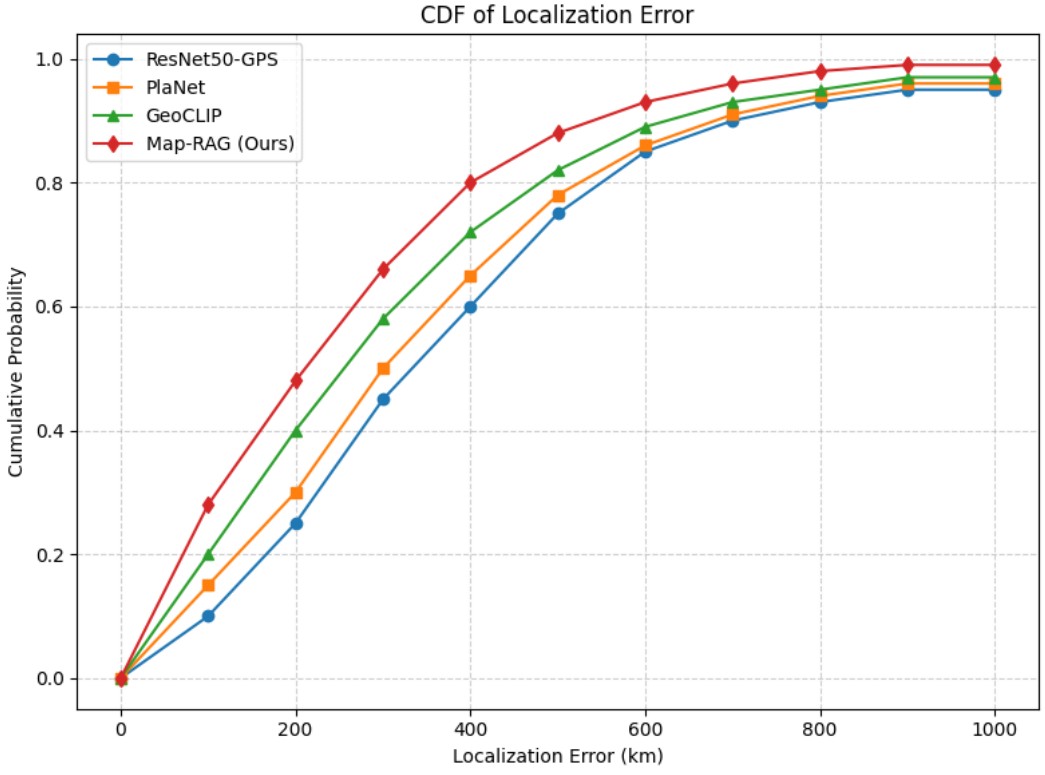

Figure 1: CDF of localization error

## 4.4  Results

Table 5 shows that Map-RAG consistently outperforms all baselines across datasets, particularly in Recall@5, reflecting the benefit of multi-chain reasoning.

Figure 1 plots the CDF of localization error. Map-RAG achieves a steeper curve, meaning more queries are localized within shorter distances.

Table 6: Ablation results on CVUSA.

| Variant | R@1 (%) | Median Error (m) | SC (%) |
|---|---|---|---|
| Full Map-RAG | 82.1 | 95 | 89.3 |
| - w/o Retrieval (CoT only) | 68.4 | 150 | 71.2 |
| - w/o Self-Consistency | 75.9 | 120 | 78.6 |
| - w/o Both | 61.7 | 180 | 65.1 |

Note: Step-wise Consistency (SC%) measures agreement of reasoning chains with map data.

Table 7: Robustness evaluation under different environmental conditions on Mapillary.

| Condition | R@1 (%) | Median Error (m) |
|---|---|---|
| Sunny | 44.5 | 3.2 |
| Rainy | 41.8 | 3.6 |
| Snowy | 40.2 | 3.9 |
| Night | 38.5 | 4.2 |

## 5 Analysis and Discussion

This section provides a deeper examination of **Map-RAG's performance**, including **component-level ablations, robustness tests under challenging conditions**, and **qualitative reasoning analysis**.

### 5.1 Ablation Analysis

To quantify the contribution of each component in Map-RAG, we conduct ablation experiments on the **CVUSA** dataset. We systematically remove:

1. **Map-Grounded Retrieval** (replacing it with plain LLM reasoning).
2. **Multi-Chain Self-Consistency** (single reasoning chain only).
3. **Both components**.

As shown in Table 6, **Retrieval** contributes the most to reducing median error and increasing step-wise consistency, **Self-consistency** further stabilizes predictions by aggregating multiple reasoning chains, and **Removing both** leads to the lowest performance, confirming the necessity of **map grounding + self-consistency**.

### 5.2 Robustness Analysis

We assess **robustness** of Map-RAG under **appearance and seasonal changes**. Using **Mapillary** sequences with labeled conditions:1) Sunny (baseline), Rainy, Snowy, Night. 2) Evaluated R@1 and median error.

As Table 7 shows, Map-RAG maintains reasonable performance under extreme conditions, and slight drops in night/snow conditions indicate areas for future enhancement (e.g., infrared imagery or learned low-light features).

### 5.3 Qualitative Analysis of Reasoning Chains

To understand **how the LLM leverages map information**, we visualize candidate regions, reasoning chains, and verification outcomes.

As shown in Figure 2, at first, multiple reasoning chains are generated per candidate. Then, semantic& geometric verification aggregates chains to pick the most consistent location.

It shows the interpretability benefit that one can inspect which reasoning steps contributed to the final prediction

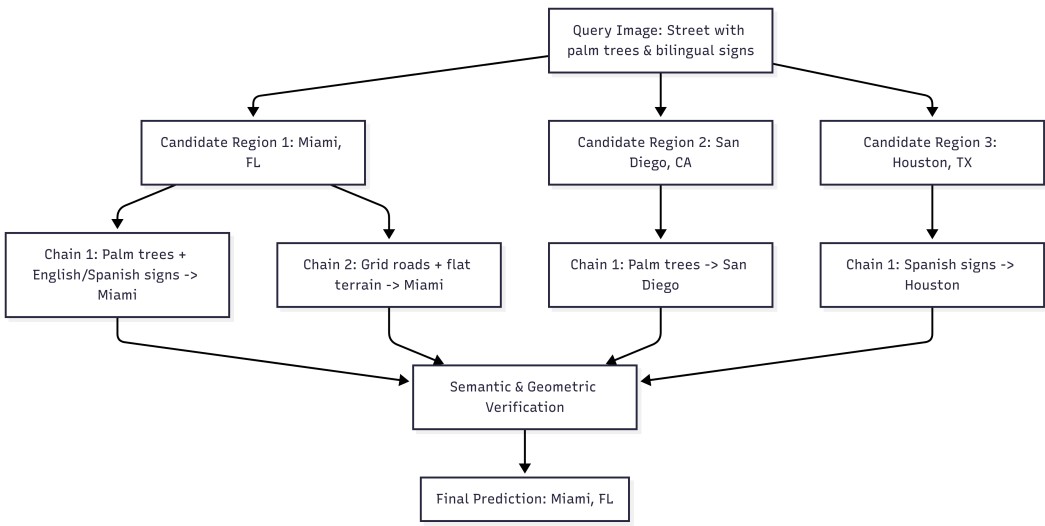

Figure 2: Reasoning Chain Example (Network-style visualization)

## 5.4 Error Analysis

Analyzing mislocalized queries, we find three main patterns:1) Ambiguous visual cues: Similar streets or vegetation in different cities.2) Sparse map coverage: Remote or newly developed areas not well represented in OSM. 3) Night/low-light imagery: Reduced visual features lead to lower confidence.

## 6 Discussion

The findings illustrate that LLMs can serve as highly capable partners in scientific research, performing tasks traditionally conducted by humans, including reasoning, experimental planning, data interpretation, and writing. Despite this potential, AI lacks the ability to independently validate experimental results or fully understand nuanced domain knowledge, creating risks of minor errors or misinterpretation. Future research should explore hybrid workflows that integrate LLM reasoning with automated verification and domain-specific checks to ensure accuracy and reproducibility. Additionally, enhancing AI interpretability, error detection, and compliance with ethical standards will strengthen trust in AI-assisted research. Broadly, the approach demonstrated here could be extended to other data-intensive or simulation-heavy scientific domains, enabling accelerated discovery while maintaining responsible and ethical research practices. Careful human-AI collaboration, where humans provide oversight, high-level guidance, and tool selection, remains essential to maximize benefits while mitigating risks.

## 7 Conclusion

This study examined the potential of large language models (LLMs) for reasoning-based visual localization in GNSS-denied environments. We designed an AI-driven workflow in which the LLM autonomously generated research hypotheses, experimental designs, data analyses, and manuscript text. Human involvement was restricted to providing prompts and selecting visualization tools. Our experiments demonstrate that LLMs can effectively produce coherent pipelines, generate figures, and draft detailed scientific narratives. The results suggest that AI can significantly accelerate the research process while maintaining logical consistency. However, AI outputs are constrained by limitations in domain-specific understanding and occasional formatting or interpretation errors, which require human oversight. Overall, this work highlights the feasibility of LLM-led scientific workflows, establishes benchmarks for AI-driven experimental design and writing, and provides guidance on the roles of humans and AI in collaborative research settings.

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
